# Piltunines A–F from the Marine-Derived Fungus *Penicillium piltunense* KMM 4668

**DOI:** 10.3390/md17110647

**Published:** 2019-11-18

**Authors:** Shamil Sh. Afiyatullov, Olesya I. Zhuravleva, Alexandr S. Antonov, Elena V. Leshchenko, Mikhail V. Pivkin, Yuliya V. Khudyakova, Vladimir A. Denisenko, Evgeny A. Pislyagin, Natalya Yu. Kim, Dmitrii V. Berdyshev, Gunhild von Amsberg, Sergey A. Dyshlovoy

**Affiliations:** 1G.B. Elyakov Pacific Institute of Bioorganic Chemistry, Far Eastern Branch of the Russian Academy of Sciences, Prospect 100-letiya Vladivostoka, 159, Vladivostok 690022, Russia; alexanderantonovpiboc@gmail.com (A.S.A.); bykadorovachem@gmail.com (E.V.L.); oid27@mail.ru (M.V.P.); vladenis@pidoc.dvo.ru (V.A.D.); pislyagin@hotmail.com (E.A.P.); natalya_kim@mail.ru (N.Y.K.);berdyshev@piboc.dvo.ru (D.V.B.); dyshlovoy@gmail.com (S.A.D.); 2School of Natural Science, Far Eastern Federal University, Sukhanova St., 8, Vladivostok 690000, Russia; 3Laboratory of Experimental Oncology, Department of Oncology, Hematology and Bone Marrow Transplantation with Section Pneumology, Hubertus Wald-Tumorzentrum, University Medical Center Hamburg-Eppendorf, 20246 Hamburg, Germany; g.von-amsberg@uke.de; 4Martini-Klinik Prostate Cancer Center, University Hospital Hamburg-Eppendorf, 20246 Hamburg, Germany

**Keywords:** *Penicillium piltunense*, secondary metabolites, carotane sesquiterpenoids, cytotoxic activity

## Abstract

Six new carotane sesquiterpenoids piltunines A–F (**1**–**6**) together with known compounds (**7**–**9**) were isolated from the marine-derived fungus *Penicillium piltunense* KMM 4668. Their structures were established using spectroscopic methods. The absolute configurations of **1**–**7** were determined based on circular dichroism (CD) and nuclear Overhauser spectroscopy (NOESY) data as well as biogenetic considerations. The cytotoxic activity of some of the isolated compounds and their effects on regulation of reactive oxygen species (ROS) and nitric oxide (NO) production in lipopolysaccharide-stimulated macrophages were examined.

## 1. Introduction

Species of the genus *Penicillium* can grow on many different substrates and they are the most widespread fungi on our planet. These fungi are commonly isolated from terrestrial substrates and are also known as permanent components of marine ecosystems. *Penicillium* species are frequently found associated with sea grasses, algae, vertebrates, and invertebrates. Among microscopic fungi, divaricate *Penicillium* species are one of the most difficult to identify. This group comprises species with asymmetric, strongly divaricate penicillia bearing terminal and subterminal clusters of monoverticillate structures. Morphological, physiological and molecular analyses of the *Penicillium* spp. isolates of mycobiota in the northeastern part of the Sakhalin Island shelf (near the Piltun Bay, baymouth) [1,2] revealed that the strain *Penicillium piltunense* KMM 4668 isolated from subaqueous soilswas not assignable to any previously known species. Phylogenetic analysis showed that the strain was more closely related to members of the *Penicillium canescens* group including eight species: *P. canescens*, *P. antarcticum*, *P. atrovenetum*, *P. coralligerum*, *P. novae-zeelandiae*, *P. jensenii*, *P. murcianum*, and *P. janczewski* and was described as a new species [3].

Fungi of the *P. canescens* group are known to produce metabolites belonging to various chemical classes: aromatic polyketide dimers canescones A–E [4], brominated azaphilones [5], tetrapeptide D-Phe-L-Val-D-Val-L-Tyr [6], isopentylateddibenzodioxocinone and pyran-3,5-dione derivatives [7], polyketides antarones A and B [8] and cladomarine [9]. These compounds exhibit cytotoxic [5], enzyme inhibitory [4], antimicrobial [9] and antifungal [6] activities.

In our search for fungal secondary metabolites with novel chemical structures and/or cytotoxic activity we have investigated the strain *Penicillium piltunense* KMM 4668. We report herein the isolation, structure determination, and biological assay results of the new carotane sesquiterpenoids piltunines A–F (**1**–**6**), known penigrisacid D (**7**) [10], cladosporin (**8**) [11] and 5’-hydroxyasperentin (**9**) [12] produced by this fungus.

## 2. Results and Discussion

The fungus was cultured for 21 days on the solid rice medium. The EtOAc extract of the mycelium was purified by a combination of Si gel and Gel ODS-A column chromatography and RP HPLC to yield compounds **1**–**9** (Figure 1) as amorphous solids.

The molecular formula of **1** was determined as C_15_H_22_O_5_ from the HRESIMS peak at *m/z* 281.1392 [M – H]^−^ and was in accordance with the ^13^C nuclear magnetic resonance (NMR) data. The ^1^H and ^13^C NMR (Table 1; Table 2; Appendix A), DEPT and HSQC spectra showed the presence of a methyl group (δ_H_ 1.81, δ_C_ 17.3), seven methylenes (δ_C_ 25.0, 32.6, 35.7, 35.9, 37.4, 66.5 and 77.0) including two oxygen-bearing, two methines (δ_H_ 2.37, 4.32, δ_C_ 57.6, 85.5) including one methine linked to an oxygen atom, one sp^3^ quaternary carbon (δ_C_ 54.4), one oxygenated quaternary carbon (δ_C_ 80.0), one tetrasubstituted double bond (δ_C_ 129.7 and 141.3) and one carbonyl or carboxyl carbon (δ_C_ 178.4).

The correlations observed in the COSY and HSQC spectra of **1** indicated the presence of the following isolated spin systems: –CH_2_–CH< (C-1–C-2), –CH_2_–CH_2_–CH< (C-4–C-6) and –CH_2_–CH_2_– (C-8–C-9). The HMBC correlations (Figure 2) from H-1β (δ_H_ 2.19) to C-2 (δ_C_ 85.5), C-3 (δ_C_ 80.0), C-6 (δ_C_ 57.6), C-7 (δ_C_ 54.4) and C-8 (δ_C_ 35.9), from H-4β (δ_H_ 2.44) toC-2, C-3, C-5 (δ_C_ 25.0) and C-6, from H-8β (δ_H_ 1.53) to C-6, C-7, C-9 (δ_C_ 32.6) and C-10 (δ_C_ 141.3) revealed the presence of a carotane bicyclic ring system. The long-range correlations from H_3_-15 (δ_H_ 1.81) to C-10, C-11 (δ_C_ 129.7) and C-14 (δ_C_ 66.5) indicated the presence of the 2-hydroxy-1-methylethylidene residue at C-10 in **1**. TheHMBC correlations from H-1α (δ_H_ 2.05), H-2 (δ_H_ 4.32) and H-8 to C-13 (δ_C_ 77.0) and downfield chemical shifts of C-2, C-3 and C-13 suggested the presence of hydroxy group at C-3 and confirmed the existence of the ether bridge between C-2 and C-13 in **1**.The treatment of **1** with diazomethane resulted in the methylated derivative **1a** as the only product. The long-range correlations from H_2_-4 (δ_H_ 1.49, 2.37) and H_3_-16 (δ_H_ 3.79) to C-12 (δ_C_ 175.9) (Figure 2, S27) revealed the presence of a carboxyl group at C-3 in **1**. Thus, the planar structure of **1** was established (Figure 2) and was very similar to aspterricacid [13] with the exception of presence of a 2-hydroxy-1-methylethylidene group at C-10 instead of an isopropylidene one.

The nuclear Overhauser spectroscopy(NOESY) correlations (Figure 3) H-6/H-1α, H-4α (δ_H_ 1.43), H-8α (δ_H_ 1.71), H-13a (δ_H_ 3.37)/H-1β, H-8β and H-9β (δ_H_ 2.11), H-13b (δ_H_ 3.73)/H-5β (δ_H_ 1.72), H_3_-15 (δ_H_ 1.81)/H-5α (δ_H_ 2.44), H-6 (δ_H_ 2.37) and H_2_-14 (δ_H_ 3.92, 4.01)/H-9α (δ_H_ 2.47) were assigned relative configuration of **1** including *E* configuration of the 10, 11 double bond. Compound **1** exhibited a nearly identical circular dichroism (CD) spectrum (Appendix A) to that of aspterric acid [10], for which the absolute configuration was established using X-ray analysis [13]. Thus, the absolute configuration of **1** was assigned as 2*R*,3*R*,6*S*,7*S*. Compound **1** was named piltunine A.

A database search revealed that compound **1** is a synthetic compound that available commercially (SciFinder.cas.org). Piltunine A (**1**) is reported here for first time as a natural product.

The HRESIMS of **2** showed the quasimolecular ion at *m/z* 323.1500 [M – H]^–^. These data, coupled with ^13^C NMR spectral data (DEPT), established the molecular formula of **2** as C_17_H_24_O_6_. The general features of ^1^H and ^13^C NMR spectra (Table 1; Table 2; Appendix A) of **2** showed a close similarity of the carbon chemical shifts to the ones for piltunine A (**1**), with the exception of the C-10, C-11 and C-14 carbon signals. The molecular mass difference of 42 mass units between **1** and **2** and HMBC correlations from H_3_-17 (δ_H_ 2.03) to C-14 (δ_C_ 69.4) and C-16 (δ_C_ 173.6) indicate the presence of an acetoxy group at C-14 in **2**.

The absolute configurations of the chiral centers in **2** were elucidated as 2*R*,3*R*,6*S*,7*S* based on NOESY data (Appendix A, Appendix A) and biogenetic considerations. Compound **2** was named piltunine B.

The molecular formula of compound **3** was determined as C_15_H_22_O_5_ based on the analysis of HRESIMS (*m/z* 281.1388 [M – H]^−^, calcd for C_15_H_22_O_5_ 281.1394) and NMR data (Table 1; Table 2; Appendix A). The ^13^C NMR data for this compound were similar to those obtained for aspterric acid [10] and aspterric acid methyl ester [14] with the exception of the C-1, C-7, C-8 and C-13 carbon signals. The molecular mass difference of 16 mass units between **3** and aspterric acid and HMBC correlations from H-1α (δ_H_ 1.89), H-2 (δ_H_ 4.40), H-8α (δ_H_ 1.46) to C-13 (δ_C_ 101.1) revealed the presence of a hydroxy group at C-13 in **3**. The absolute configurations of the chiral centers in **3** were elucidated as 2*R*,3*R*,6*S*,7*S*,13*R* based on NOESY data (Figure 3) and biogenetic considerations. Compound **3** was named piltunine C.

(–)HRESIMS **4** gave a quasimolecular ion at *m/z* 295.1548 [M – H]^−^. These data, coupled with ^13^C NMR spectroscopic data ( Table 1; Table 2; Appendix A), established the molecular formula of **4** as C_16_H_24_O_5_. The structures of the carotane seven-membered and furan rings in **4** were determined by HMBC correlations (Appendix A) as for piltunine A (**1**). The long-range correlations from H-8β (δ_H_ 2.06) to C-6 (δ_C_ 60.4), C-7 (δ_C_ 56.0), C-9 (δ_C_ 129.7) and C-10 (δ_C_ 152.0), from H-9 (δ_H_ 5.68) to C-10, C-11 (δ_C_ 77.7), from H_3_-14 (δ_H_ 1.30) to C-10, C-11, C-15 (δ_C_ 26.6) and from H_3_-16 (δ_H_ 3.12) to C-11 elucidated the structures of a five-membered ring with ∆^9,10^ double bond and side chain, including the location of the tertiary methoxy group at C-11.

The relative configuration of **4** was assigned based on NOESY correlations (Appendix A) H-6 (δ_H_ 2.09)/H-1α (δ_H_ 2.16), H-4α (δ_H_ 1.37) and H-8β (δ_H_ 2.06)/H-1β (δ_H_ 2.21), H-13a (δ_H_ 3.38) and H-13b (δ_H_ 3.95)/H-5β(δ_H_ 1.88). The absolute configurations of the chiral centers in **4** were defined to be the same as that of piltunine A (**1**) based on biogenetic consideration. Compound **4** was named piltunine D.

The HRESIMS of **5** showed the quasimolecular ion at *m/z* 265.1077 [M – H]^–^. These data, coupled with ^13^C NMR spectral data (Table 1 and Table 2; Appendix A), established the molecular formula of **5** as C_14_H_18_O_5_. The ultraviolet (UV) spectrum exhibits a λ_max_ at 239 nm (logε3.76) and at 195 nm (logε 3.50), consistent with the enone system in **5**. The structures of the carotane seven-membered and furan rings in **5** were determined by HMBC correlations (Appendix A) as for piltunines A (**1**) and D (**4**). The HMBC correlations from H-8β (δ_H_ 2.62) to C-6 (δ_C_ 58.7), C-7 (δ_C_ 55.5), C-9 (δ_C_ 147.8) and C-10 (δ_C_ 149.0), from H-9 (δ_H_ 6.90) to C-6, C-7, C-10, C-11 (δ_C_ 200.8), from H_3_-14 (δ_H_ 2.26) to C-10 and C-11 elucidated the structures of a five-membered ring and indicated the 9-en-11-one position for the trisubstituted enonechromophore in **5**. The absolute configurations of the chiral centers in **5** were elucidated based on a NOESY experiment (Appendix A) and biogenetic considerations as for piltunine D (**4**). Compound **5** was named piltunine E.

The molecular formula of **6** was determined as C_14_H_18_O_5_ based on the analysis of HRESIMS (*m/z* 265,1080 [M–H]^−^, calcd for C_14_H_17_O_5_ 265,1081) and by ^13^C NMR analyses (Table 1 and Table 2; Appendix A).The UV spectrum exhibits a λ_max_ at 255 nm (logε4.07) and at 196 (logε 3.88), consistent with the enone system in **6**. The HMBC correlations (Appendix A) from H-1β (δ_H_ 2.07) to C-2 (δ_C_ 85.0), C-3 (δ_C_ 78.6), C-6 (δ_C_ 162.7), C-7 (δ_C_ 60.5) and C-8 (δ_C_ 33.4), from H-4β (δ_H_ 2.26) to C-2, C-3, C-5 (δ_C_ 23.3), C-6 and C-12 (δ_C_ 178.3), from H_2_-8 (δ_H_ 1.89, 1.96) to C-1 (δ_C_ 39.1), C-6, C-7, C-9 (δ_C_ 33.7), C-10 (δ_C_ 135.4) and C-13 (δ_C_ 79.7)and from H_3_-14 (δ_H_ 2.24) to C-9, C-10 and C-11 (δ_C_ 202.5) revealed the presence of a carotane bicyclic ring system, an ester bridge between C-2 and C-13 and 6-en-11-one position for the tetrasubstituted enone chromophore in **6**. The absolute configurations of the chiral centers in **6** were defined based on NOESY correlations H-13a (δ_H_ 3.74)/H-1β (δ_H_ 2.07), H-8β(δ_H_ 1.89), H-13b (δ_H_ 3.84)/H-5β(δ_H_ 2.74), H-1α (δ_H_ 2.32)/H-4α (δ_H_ 1.46) (Appendix A) and biogenetic considerations as 2*R*,3*R*,7*S*. Compound **6** was named piltunine F.

The structures of known compounds penigrisacid D (**7**) [10] (Appendix A, Appendix A), cladosporin (**8**) (Appendix A) [11] and 5’-hydroxyasperentin (**9**) (Appendix A) [12] were determined on the basis of HRESIMS and NMR data as well ascomparison with literaturedata.

The absolute configurations of the chiral centers in penigrisacid D (**7**)were defined based on NOESY (Figure 4, Appendix A) correlations H-6 (δ_H_ 1.71)/H-1α (δ_H_ 2.03), H-4α (δ_H_ 1.41), H-8α (δ_H_ 1.85), H-15b (δ_H_ 5.01), H-13a (δ_H_ 3.41)/H-1β (δ_H_ 2.25), H-8β(δ_H_ 1.89), H-13b (δ_H_ 4.67)/H-5β(δ_H_ 1.81), H-9β (δ_H_ 1.65) and H-9α (δ_H_ 2.12)/H_3_-14 (δ_H_ 1.78) and biogenetic considerations as 2*R*,3*R*,6*S*,7*S*,10*S*.

We have investigated the effect of the isolated compounds **1**, **1a**, **2**, **7**–**9** on the viability of human drug-resistant prostate cancer 22Rv1 cell as well as on human prostate non-cancer PNT-2 cells using an MTT assay (Table 3). We have also determined a selectivity index (SI) at for all the tested substances (Table 3). Compounds showing high SI values are more active in cancer cells in comparison with non-cancer cells lines. Thus, **8** had the highest SI value among the tested substances, which makes it a promising candidate for the further synthetic modifications in order to optimize its anticancer activity and selectivity. Interestingly, **8** has been also identified to be the most cytotoxic in cancer cells in comparison with the other substances (Table 3). Note, 22Rv1 cells are known to be resistant to the hormone therapy due to the expression of androgen receptor splice variant AR-V7 [15]. Thus, the compounds which are active inAR-V7-positive 22Rv1 cells are of high clinical interest.

Additionally, we have investigated the effects of the isolated compounds **1**, **2**, **3** and **5**–**9** on reactive oxygen species (ROS) and nitric oxide (NO) production in murine macrophages following lipopolysaccharide (LPS) stimulation. NO, which is produced in large quantities by inducible nitric oxide synthase (iNOS), is known to be responsible for vasodilation and hypotension observed during septic shock and inflammation. 

Compounds **5** and **9** induced a significant down-regulation of ROS production (Figure 5). LPS from E. coli, an inflammatory agent, was used as a positive control in our study.

Compound **9** suppressed NO production in LPS-stimulated macrophages at non-toxic concentration of 1 μM. NO level in these cells was decreased by 24.1 ± 0.7% (*p* < 0.05, Student’s *t*-test) in comparison with LPS-treated control (data not shown). Therefore, compound **9** may be a promising candidate for the therapy of inflammatory diseases accompanying overproduction of NO.

## 3. Materials and Methods

### 3.1. General Experimental Procedures

Optical rotations were measured on a Perkin-Elmer 343 polarimeter (Perkin Elmer, Waltham, MA, USA). UV spectra were recorded on a Shimadzu UV-1601PC spectrometer (Shimadzu Corporation, Kyoto, Japan) in methanol. CD spectra were measured with a Chirascan-Plus CD Spectrometer (Leatherhead, UK) in methanol. NMR spectra were recorded in CD_3_OD and CDCl_3_, on a Bruker DPX-500 (Bruker BioSpin GmbH, Rheinstetten, Germany) and a Bruker DRX-700 (Bruker BioSpin GmbH, Rheinstetten, Germany) spectrometer, using TMS as an internal standard. HRESIMS spectra were measured on a Maxis impact mass spectrometer (Bruker Daltonics GmbH, Rheinstetten, Germany).

Low-pressure liquid column chromatography was performed using Si gel L (50/100 μm, Imid, Russia) and Gel ODS-A (12 nm, S – 75 um, YMC Co, Ishikawa, Japan). Plates precoated with Si gel (5–17 μm, 4.5 × 6.0 cm, Imid) and Si gel60 RP-18 F_254_S (20 × 20 cm, Merck KGaA, Germany) were used for thin-layer chromatography. Preparative HPLC was carried out on aAgilent 1100 chromatograph (Agilent Technologies, USA) using a YMC ODS-AM (YMC Co., Ishikawa, Japan) (5 µm, 10 × 250 mm), YMC ODS-A (YMC Co., Ishikawa, Japan) (5 µm, 4.6 × 250 mm) and Supelco Discovery C-18 (5 μm, 250 × 4.6 mm) columns with a Agilent 1100 refractometer (Agilent Technologies, Santa Clara, CA, USA).

### 3.2. Fungal Strain

The strain of Penicillium piltunense KMM 4668 was isolated from marine subaqueous soil (Piltun Bay, Sea of Okhotsk, the northeastern shelf of the Sakhalin Island, Russia). The fungus was identified according to a molecular biology protocol by DNA amplification and sequencing of the ITS region (MycoBank accession number is MB818671) and morphological and physiological studies. The strain is stored at the Collection of Marine Microorganisms (KMM) of G.B. Elyakov Pacific Institute of Bioorganic Chemistry (Vladivostok, Russia).

### 3.3. Cultivation of Fungus

The fungus was cultured at 22 °C for three weeks in 60 × 500 mL Erlenmeyer flasks, each containing rice (20.0 g), yeast extract (20.0 mg), KH_2_PO_4_ (10 mg) and natural sea water from the Marine Experimental Station of PIBOC, Troitsa (Trinity) Bay, Sea of Japan (40 mL).

### 3.4. Extraction and Isolation

At the end of the incubation period, the mycelia and medium were homogenized and extracted with EtOAc (1 L). The obtained extract was concentrated to dryness. The residue (4.8 g) was dissolved in H_2_O−EtOH (4:1) (100 mL) and was extracted with *n*-hexane (0.2 L × 3) and EtOAc (0.2 L × 3). After evaporation of the EtOAc layer, the residual material (2.5 g) was passed over a silica column (3 × 14 cm), which was eluted first with *n*-hexane (200 mL) followed by a step gradient from 5% to 50% EtOAc in *n*-hexane (total volume 20 L). Fractions of 250 mL were collected and combined on the basis of TLC (Si gel, toluene–isopropanol 6:1 and 3:1, v/v).

The *n*-hexane–EtOAcfraction (80:20, 570 mg) wasseparated on a Gel ODS-A column (1.5 × 8 cm), which was eluted by a step gradient from 40% to 80% MeOH in H_2_O (total volume 1 L) to yield subfractions I, II and III. Subfraction I (40% MeOH, 100 mg) was purified by RP HPLC on a YMC ODS-A column eluting with MeOH–H_2_O (40:60) to yield **1** (15 mg). Subfraction II (60% MeOH, 143 mg) was purified by RP HPLC on a YMC ODS-AM column eluting at first with MeOH–H_2_O–TFA (65:35:0.1) and then with MeOH–H_2_O (55:45) to yield **2** (13 mg), **7** (7 mg), **9** (7.5 mg). Subfraction III (80% MeOH, 60 mg) was purified and separated by RP HPLC on a YMC ODS-A column eluting at first with MeOH–H_2_O (80:20) and then with MeOH–H_2_O (60:40) to yield **8** (10 mg) and **4** (1 mg).

The *n*-hexane–EtOAcfraction (70:30, 70 mg) was separated on a Gel ODS-A column (1.5 × 8 cm) eluting with MeOH–H_2_O (40:60) and then was purified on an YMC ODS-A column eluting with MeOH–H_2_O (60:40) to yield **3** (3 mg).

The *n*-hexane–EtOAcfraction (50:50, 500 mg) was separated on a Gel ODS-A column (1.5 × 8 cm) eluting with MeOH–H_2_O (60:40) and then was purified on a Supelco-C-18 column eluting with MeOH–H_2_O (35:65) to yield **5** (6 mg) and **6** (4.6 mg).

### 3.5. Spectral Data

Piltunine A (**1**): amorphous solids; [α]_D_^20^–91.6 (*c* 0.1 CH_3_OH); UV (CH_3_OH) λ_max_ (log ε) 242 (3.19) and 203 (3.73) nm, see Appendix A; CD (*c* 2.1 × 10^−3^, CH_3_OH), λ_max_(∆ε) 207 (+4.61), 229 (–0.29), 250 (+0.45) nm, see Appendix A; infrared (IR) (CDCl_3_) ν_max_ 3500, 2947, 1751, 1714, 1670, 1485, 1412, 1390, 1216, 1091, 1048 cm^−1^, see Appendix A; ^1^H and ^13^C NMR data, see Table 1 and Table 2, Appendix A; HRESIMS *m*/*z* 281.1392 [M – H]^–^ (calcd. for C_15_H_21_O_5_, 281.1394, Δ +1.0 ppm), 305.1352 [M + Na]^+^ (calcd. for C_15_H_22_O_5_Na, 305.1359, Δ +2.6 ppm).

Piltunine B (**2**): amorphous solids; [α]_D_^20^–55 (*c* 0.09 CH_3_OH); UV (CH_3_OH) λ_max_ (log ε) 250 (3.14) and 201 (3.80) nm, see Appendix A; CD (*c* 2.2 × 10^−3^, CH_3_OH), λ_max_(∆ε) 209 (+0.45), 225 (−0.51), 252 (+0.28) nm, see Appendix A; IR (CDCl_3_) ν_max_ 3500, 2937, 1758, 1744, 1714, 1602, 1385, 1241, 1051 cm^−1^, see Appendix A; ^1^H and ^13^C NMR data, see Table 1 and Table 2, Appendix A; HRESIMS *m*/*z* 323.1500 [M – H]^–^ (calcd. for C_17_H_23_O_6_, 323.1500, Δ +0.1 ppm), 347.1468 [M + Na]^+^ (calcd. for C_17_H_24_O_6_Na, 347.1465, Δ −0.9 ppm).

Piltunine C (**3**): amorphous solids; [α]_D_^20^–145 (*c* 0.1 CH_3_OH); UV (CH_3_OH) λ_max_ (log ε) 199 (3.66) nm, see Appendix A; CD (*c* 2.1 × 10^−3^, CH_3_OH), λ_max_(∆ε) 204 (+4.77), 226 (–0.33) nm, see Appendix A; ^1^H and ^13^C NMR data, see Table 1 and Table 2, Appendix A; HRESIMS *m*/*z* 281.1388 [M – H]^–^ (calcd. for C_15_H_21_O_5_, 281.1394, Δ +2.2 ppm), 305.1355 [M + Na]^+^ (calcd. for C_15_H_22_O_5_Na, 305.1359, Δ +1.6 ppm).

Piltunine D (**4**): amorphous solids; [α]_D_^20^–25 (*c* 0.08 CH_3_OH); UV (CH_3_OH) λ_max_ (log ε) 199 (3.55) nm, see Appendix A; CD (*c* 2.7 × 10^−3^, CH_3_OH), λ_max_(∆ε) 221 (−0.48), 250 (+0.24), 297 (−0.05) nm; IR (CDCl_3_) ν_max_ 2929, 2857, 1745, 1713, 1457, 1216, 1068 cm^−1^, see Appendix A; ^1^H and ^13^C NMR data, see Table 1 and Table 2, Appendix A; HRESIMS *m*/*z* 295.1548 [M – H]^–^ (calcd. for C_16_H_23_O_5_, 295.1551, Δ +0.9 ppm).

Piltunine E (**5**): amorphous solids; [α]_D_^20^–84.5 (*c* 0.10 CH_3_OH); UV (CH_3_OH) λ_max_(log ε) 239 (3.76) and 195 (5.50) nm, see Appendix A; CD (*c* 3.0 × 10^−3^, CH_3_OH), λ_max_(∆ε) 195 (+6.25), 238 (−1.16) nm, see Appendix A; ^1^H and ^13^C NMR data, see Table 1 and Table 2, Appendix A; HRESIMS *m*/*z* 265.1077 [M – H]^–^ (calcd. for C_14_H_17_O_5_, 265.1081, Δ +1.7 ppm), 289.1043 [M + Na]^+^ (calcd. for C_14_H_18_O_5_Na, 289.1046, Δ +1.1 ppm).

Piltunine F (**6**): amorphous solids; [α]_D_^20^ −16.1 (*c* 0.14 CH_3_OH); UV (CH_3_OH) λ_max_(log ε) 255 (4.07) and 196 (3.88) nm, see Appendix A; CD (*c* 1.2 × 10^−3^, CH_3_OH), λ_max_(∆ε) 214 (−1.65), 257 (+3.86) nm, see Appendix A; IR (CDCl_3_) ν_max_ 3500, 2945, 1759, 1673, 1605, 1385, 1259, 1209, 1136, 1112, 1053 cm^−1^, see Appendix A; ^1^H and ^13^C NMR data, see Table 1 and Table 2, Appendix A; HRESIMS *m*/*z* 265.1080 [M – H]^–^(calcd. for C_14_H_17_O_5_, 265.1081, Δ +0.4 ppm), 289.1042 [M + Na]^+^ (calcd. for C_14_H_18_O_5_Na, 289.1046, Δ +1.4 ppm).

Penigrisacid D (**7**): amorphous solids; [α]_D_^20^–36.2 (*c* 0.16 EtOH); CD (*c* 2.7 × 10^−3^, CH_3_OH), λ_max_(∆ε) 209 (−1.03), 257 (+0.03), 300 (–0.12) nm, see Appendix A; ^1^H and ^13^C NMR data, see Appendix A; HRESIMS *m*/*z* 281.1394 [M – H]^–^ (calcd. for C_15_H_21_O_5_, 281.1394, Δ +0.3 ppm), 305.1358 [M + Na]^+^ (calcd. for C_15_H_22_O_5_Na, 305.1359, Δ +0.5 ppm).

Compounds **1a**: amorphous solids; ^1^H and ^13^C NMR data, see Appendix A; HRESIMS *m*/*z* 319.1512 [M + Na]^+^ (calcd. for C_16_H_24_O_5_Na, 319.1516, Δ +1.2 ppm).

### 3.6. Cell Lines and Culture Conditions

The human prostate cancer cells 22Rv1 and human prostate non-cancer cells PNT-2 were purchased from ATCC. Cell lines were cultured according to the manufacturers instructions in 10% FBS/RPMI media (Invitrogen). Cells were continuously kept in culture for a maximum of 3 months, and were routinely inspected microscopically for stable phenotype and regularly checked for contamination with mycoplasma. The authentity of the cells has recently been confirmed by the Multiplexion (Heidelberg, Germany) using highly polymorphic short tandem repeat loci.

### 3.7. In Vitro MTT-Based Drug Sensitivity Assay

The *in vitro* cytotoxicity of individual substances was evaluated using the MTT (3-(4,5-dimethylthiazol-2-yl)-2,5-diphenyltetrazolium bromide) assay, which was performed as previously described [16]. The vehicle (DMSO) treated cells were used as a negative control. Treatment time was 72 h.

### 3.8. Reactive Oxygen Species (ROS) Level Analysis in Lipopolysaccharide (LPS)-Treated Cells

The suspension of macrophages on 96-well plates (2 × 10^4^ cells/well) were washed withphosphate-buffered saline (PBS) and treated with 180 µL/well of the tested compounds (10 μM) for 1 h and 20 µL/well LPS from *E. coli* serotype 055:В5 (Sigma-Aldrich, MA, USA, 1.0 μg/mL), which were both dissolved in PBS and cultured at 37 °C in a CO_2_-incubator for one hour. For the ROS levels measurement, 200 μL of 2,7-dichlorodihydrofluorescein diacetate (DCF-DA, Sigma, final concentration 10 μM) fresh solution was added to each well, and the plates were incubated for 30 min at 37 °C. The intensity of DCF-DA fluorescence was measured at λex 485 n/λem 518 nm using the plate reader PHERAstar FS (BMG Labtech, Offenburg, Germany) [17].

### 3.9. Reactive Nitrogen Species (RNS) Level Analysis in LPS-Treated Cells

The suspension of macrophages on 96-well plates (2 × 10^4^ cells/well) were washed with the PBS and treated with 180 µL/well of the tested compounds (10 μM) for 1 h and 20 µL/well LPS from *E. coli* serotype 055:В5 (Sigma, 1.0 μg/mL), which were both dissolved in PBS and cultured at 37 °C in a CO_2_-incubator for one hour. For the RNS levels measurement, 200 μL Diaminofluorescein-FM diacetate (DAF FM-DA, Sigma, final concentration 10 μM) fresh solution was added to each well, and the plates were incubated for 40 min at 37 °C, then replace with fresh PBS, and then incubated for an additional 30 min to allow complete de-esterification of the intracellular diacetates. The intensity of DAF FM-DA fluorescence was measured at λex 485 n/λem 520 nm using the plate reader PHERAstar FS (BMG Labtech, Offenburg, Germany).

### 3.10. Peritoneal Macrophage Isolation

Mice BALB/c were sacrificed by cervical dislocation. Peritoneal macrophages were isolated using standard procedures. For this purpose, 3 ml of PBS (pH 7.4) was injected into the peritoneal cavity and the body intensively palpated for 1–2 min. Then the peritoneal fluid was aspirated with a syringe. Mouse peritoneal macrophage suspension was applied to a 96-well plate left at 37 °C in an incubator for 2 h to facilitate attachment of peritoneal macrophages to the plate. Then a cell monolayer was triply flushed with PBS (pH 7.4) for deleting attendant lymphocytes, fibroblasts and erythrocytes and cells were used for further analysis.

All animal experiments were conducted in compliance with all rules and international recommendations of the European Convention for the Protection of Vertebrate Animals used for experimental and other scientific purposes. All procedures were approved by the Animal Ethics Committee at the G. B. Elyakov Pacific Institute of Bioorganic Chemistry, Far Eastern Branch of the Russian Academy of Sciences (Vladivostok, Russia), according to the Laboratory Animal Welfare guidelines.

### 3.11. Statistical Analysis

Average value, standard error, standard deviation and *p*-values in all experiments were calculated and plotted on the chart using SigmaPlot 3.02 (Jandel Scientific, San Rafael, CA, USA) or GraphPad Prism software v. 5.01 (GraphPad Prism software Inc., La Jolla, CA, USA). Statistical difference was evaluated by the *t*-test, and results were considered as statistically significant at *p* < 0.05.

## 4. Conclusions

Six carotane sesquiterpenoids piltunines A–F (**1**–**6**) together with known compounds (**7**–**9**) were isolated from the fungus *Penicillium piltunense* KMM 4668 obtained from marine subaqueous soils. The absolute configurations of **1**–**7** were determined using combined CD and NOESY dataas well as biogenetic considerations. Piltunine C (**3**) was established as a 13-hydroxy derivative of aspterric acid. Piltunines E (**5**) and F (**6**) contain the 9-en-11-one and 6-en-11-one enone chromophoresin structure of molecules. Piltunine A (**1**) is reported here for first time as a natural product. Some of the isolated compounds exhibited cytotoxic activity in human drug-resistant prostate cancer cells as well as being able to inhibitROS and NO production in LPS-stimulated macrophages.

## Figures and Tables

**Figure 1 marinedrugs-17-00647-f001:**
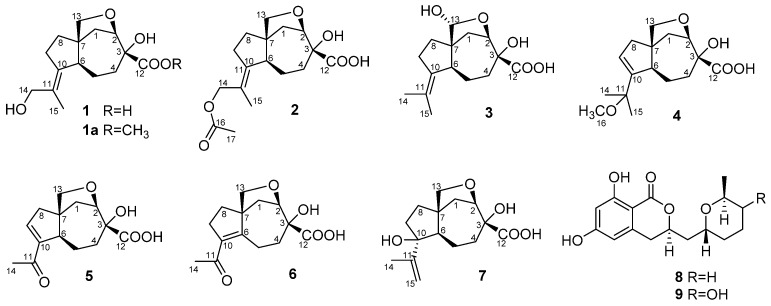
Chemical structures of **1**–**9**.

**Figure 2 marinedrugs-17-00647-f002:**
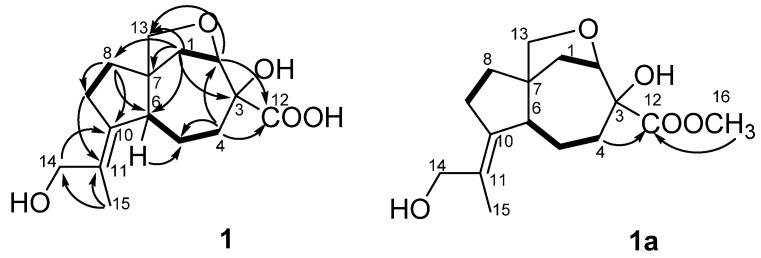
Key HMBC correlations of **1** and **1a**.

**Figure 3 marinedrugs-17-00647-f003:**
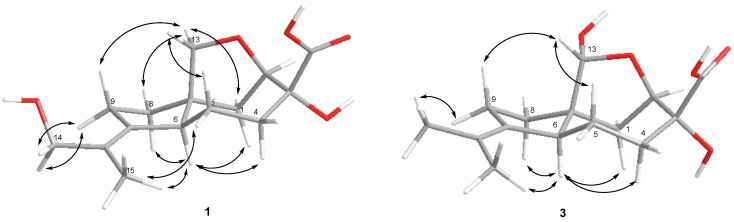
Nuclear Overhauser spectroscopy (NOESY) correlations of **1** and **3**.

**Figure 4 marinedrugs-17-00647-f004:**
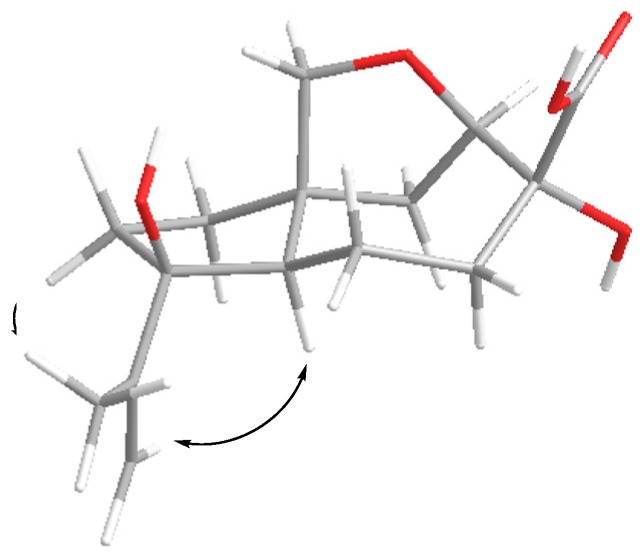
NOESY correlations of **7**.

**Figure 5 marinedrugs-17-00647-f005:**
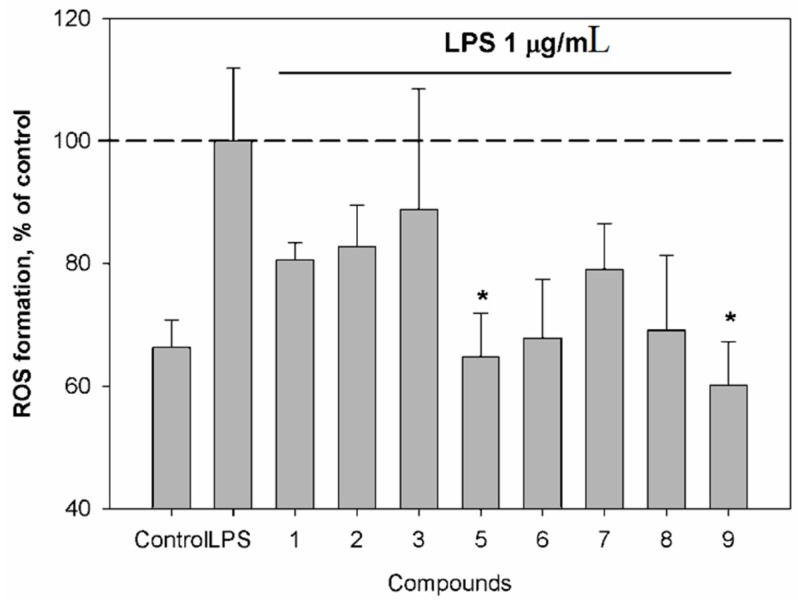
Effect of compounds on reactive oxygen species (ROS) level in murine peritoneal macrophages co-incubated with lipopolysaccharide (LPS). The compounds were tested at a concentration of 10 μM. Time of cell incubation with compounds was 1 h at 37 °C. * *p* < 0.05. (Student’s *t*-test).

**Table 1 marinedrugs-17-00647-t001:** ^1^H nuclear magnetic resonance (NMR) data (*δ* in ppm, *J* in Hz) for piltunines A–F (**1**–**6**) ^a^.

Position	1	2	3	4	5	6
1	α: 2.05 d (13.1) β: 2.19 dd (8.6, 13.3)	α: 2.05 d (14.1) β: 2.19 d (8.2)	α: 1.89 d (13.0) β: 2.28 dd (8.7, 13.0)	α: 2.16 d (13.1) β: 2.21 dd (8.1, 13.0)	α: 2.19 d (13.5) β: 2.24 d (8.8)	α: 2.32 d (13.4) β: 2.07 dd (8.3, 13.4)
2	4.32 d (8.5)	4.32 d (8.1)	4.40 d (8.5)	4.33 d (8.5)	4.36 d (8.4)	4.42 d (8.4)
4	α: 1.43 t (12.0) β: 2.44 m	α: 1.43 t (11.0) β: 2.44 m	α: 1.42 t (12.3) β: 2.43 m	α: 1.37 m β: 2.46 m	α: 1.39 t (14.0) β: 2.48 dd (8.0, 14.0)	α: 1.46 ddd (14.7, 10.4, 1.5) β: 2.26 dd (10.3, 14.6)
5	α: 2.44 m β: 1.72 m	α: 2.43 m β: 1.73 m	α: 2.40 m β: 1.71 dd (14.0, 12.6)	α: 2.37 dd (7.9, 14.5) β: 1.88 q (14.5)	α: 2.61 m β: 1.64 q (13.0)	α: 3.12 ddd (15.6, 9.8, 1.6) β: 2.74 ddd (15.9, 10.4, 1.7)
6	2.37 brd (11.6)	2.39 d (12.0)	2.31 d (14.0)	2.59 d (12.0)	2.70 d (13.0)	
7						
8	α: 1.71 m β: 1.53 dd (7.2, 12.1)	α: 1.74 m β: 1.54 dd (8.1, 12.0)	α: 1.46 dd (3.1, 12.2) β: 1.94 dd (7.1, 12.2)	α: 2.48 m β: 2.06 dd (2.8, 15.6)	α: 2.26 dd (2.6, 14.0) β: 2.62 m	α: 1.96 m β: 1.89 m
9	α: 2.47 m β: 2.11 m	α: 2.49 dd (9.1, 17.2) β: 2.17 m	α: 2.24 m β: 2.21 m	5.68 dd (2.8, 5.0)	6.90 q (2.6)	2.67 m
13	a: 3.37 dd (1.4, 8.3) b: 3.73 d (8.1)	a: 3.38 dd (1.0, 8.1) b: 3.71 d (8.1)	4.74 s	a: 3.38 d (8.5) b: 3.98 d (8.4)	a: 3.38 d (8.7) b: 3.91 d (8.4)	a: 3.74 d(8.8) b: 3.84 d (8.4)
14	a: 3.92 d (11.8) b: 4.01 d (11.8)	a: 4.48 d (12.0) b: 4.52 d (12.1)	1.62 s	1.30 s	2.26 s	2.24 s
15	1.81 brs	1.78 brs	1.74 brs	1.31 s		
16				3.12 s		
17		2.03 s				

^a^ Chemical shifts were measured at 700.13 Hz in CD_3_OD.

**Table 2 marinedrugs-17-00647-t002:** ^13^C nuclear magnetic resonance (NMR) data (*δ* in ppm) for piltunines A–F (**1**–**6**) ^a^.

Position	1	2	3	4	5	6
1	37.4 CH_2_	37.3 CH_2_	34.3 CH_2_	36.9 CH_2_	36.3 CH_2_	39.1 CH_2_
2	85.5 CH	85.5 CH	84.1 CH	85.6 CH	85.7 CH	85.0 CH
3	80.0 C	79.9 C	78.9 C	80.0 C	80.0 C	78.6 C
4	35.7 CH_2_	35.6 CH_2_	35.6 CH	36.7 CH_2_	36.4 CH_2_	32.6 CH_2_
5	25.0 CH_2_	25.0 CH_2_	25.3 CH_2_	23.5 CH_2_	21.8 CH_2_	23.3 CH_2_
6	57.6 CH	57.7 CH	58.1 CH	60.4 CH	58.7 CH	162.7 C
7	54.4 C	54.5 C	58.5 C	56.0 C	55.5 C	60.5 C
8	35.9 CH_2_	35.7 CH_2_	31.9 CH_2_	40.4 CH_2_	40.9 CH_2_	33.4 CH_2_
9	32.6 CH_2_	32.8 CH_2_	32.6 CH_2_	129.7 CH	147.8 CH	33.7 CH_2_
10	141.3 C	144.5 C	137.4 C	152.0 C	149.0	135.4 C
11	129.7 C	125.2 C	124.8 C	77.7 C	200.8 C	202.5 C
12	178.4 C	178.4 C	178.4 C	178.6 C	178.5 C	178.3 C
13	77.0 CH_2_	77.0 CH_2_	101.1 CH	78.5 CH_2_	77.6 CH_2_	79.7 CH_2_
14	66.5 CH_2_	69.4 CH_2_	24.3 CH_3_	27.4 CH_3_	27.9 CH_3_	31.0 CH_3_
15	17.3 CH_3_	17.5 CH_3_	21.4 CH_3_	26.6 CH_3_		
16		173.6 C		51.8 CH_3_		
17		21.4 CH_3_				

^a^ Chemical shifts were measured at 176.04 Hz in CD_3_OD.

**Table 3 marinedrugs-17-00647-t003:** Cytotoxic activity of the investigated compounds. Cells were treated for 72 h and the viability was measured using MTT assay. IC_50_s were calculated using GraphPad Prism software. Additionally, the viability of the cells treated with 100 µM of the compounds is presented and the selectivity index (SI) was calculated as [viability of PNT-2 cells]/[viability of 22Rv1 cells]. Docetaxel was used as a positive control.

Compound	IC_50_		Viability at 100 µM, % of Control		SI
	22Rv1 Cells	PNT-2 Cells	22Rv1 Cells	PNT-2 Cells	
**1**	>100 µM	>100 µM	65.5% ± 1.3%	97.2% ± 3.5%	1.48
**1a**	>100 µM	>100 µM	62.2% ± 4.7%	80.4% ± 0.5%	1.29
**2**	>100 µM	>100 µM	61.9% ± 3.6%	82.9% ± 1.5%	1.34
**7**	>100 µM	>100 µM	63.8% ± 2.5%	80.7% ± 8.1%	1.26
**8**	71.74 ± 8.9 µM	>100 µM	38.7% ± 3.4%	88.1% ± 3.1%	2.28
**9**	>100 µM	>100 µM	54.9% ± 6.9%	55.5% ± 4.1%	1.01
**Docetaxel**	2.78 ± 1.07 nM	84.74 ± 20.58 nM	-	-	-

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
