# Peer review of "Piltunines A–F from the Marine-Derived Fungus Penicillium piltunense KMM 4668"

_marinedrugs, 2019, doi:10.3390/md17110647_

Round 1

Reviewer 1 Report

This manuscript describes the isolation and structure elucidation of six new sesquiterpenes, piltunines A-F (1-6) from a marine derived fungus. The new metabolites appear to be adequately characterised, and the structure assignments acceptable – subject to comments shown below.

Piltunine A (1) and B (2). There is no discussion on how the geometry of the 10,11 double bond was determined, even though it is consistently drawn in a Z configuration.This issue needs to be addressed.

NMR Arrow Diagrams. There is no doubt that selected 2D NMR correlations can provide very informative "arrow" diagrams (ie Fig 2), but these are not a substitute for comprehensive documentation and analysis of the full 2D NMR data set. Given the potential for large numbers of 2D NMR correlations, it's important to know that all the correlations fit the proposed structure, and not simply focus on those cherry-picked for the arrow diagrams. Put differently, sometimes it's the correlations that "don't fit" the proposed structure that are most informative, rather than those that "do fit". It's also worth noting that the structure elucidation of 3-6 is highly dependent on analysis of the 2D NMR data, however, not only is the full data not tabulated in the SI, there are no arrow diagrams in the manuscript. The manuscript should include an informative arrow diagram for each of the new compounds.

NMR Tabulation. While the inclusion of images of NMR spectra in the Supporting Information can be useful, it is important that authors take the opportunity to fully analyse and tabulate all the 1D and 2D NMR data (even if only in the Supporting Information).  This is all the more important given that spectral images provided in the SI are unannotated, and the lack of key expansions makes it impossible for the reader/reviewer to check all assignments. It is also important because the Tables in the manuscript make assignments that are predicated on a full analysis of the 2D NMR data. The SI should include a fully tabulated account of all the 2D NMR data for each compound.

Assays

Table 3. The data lacks positive and negative controls, which make it impossible to assess the viability of cells, and the absolute potency of the compounds. Relative potency between cancer and normal cells lacks meaning, if none of the compounds are particularly active. Under these experimental conditions how do these cells perform when treated with a sham (ie DMSO) and how do they perform when treated with a well-established cytotoxic compound, and how does this correlate with the test compounds.

Table 4. What is the positive control?

Table 5. I am at a loss to understand the significance of this assay. Are the compounds nOs inhibitors? Are they NO quenching agents? Do they block LPS mediated signalling?

The biological assays appear to be somewhat randomly selected, with little effort made to explain their purpose and the actual results – other than cite the readout. More effort could be given to explaining

(i) why these assays were selected,

(ii) how they we actually performed, including controls,

(iii) what readout constitutes significance and WHY, and

(iv) if results are significant, what is their relevance going forward.

The manuscript does describe some interesting natural products, and the structure assignments are likely correct. Notwithstanding concerns I have over double bond configurations, the appropriate documentation of 1D and 2D NMR data, and the bioassays, the manuscript could be made suitable for publication. I would encourage the authors to take heed of the comments made above, and make appropriate adjustments.

Author Response

Dear Reviewer. Thank you for your careful analysis of our paper and valuable comments. Bellow you can find the answers for your questions and comments.

Q: Piltunine A (1) and B (2). There is no discussion on how the geometry of the 10,11 double bond was determined, even though it is consistently drawn in a Z configuration. This issue needs to be addressed.

A: A careful analysis of the observed NOE correlations shows that the hydroxy group in 1 is at C-14, and the 10,11 double bond has the E configuration. Accordingly, we have made changes in the sentence on line 85, Figures 1, 2 and 3 and Tables 1, 2. Similar corrections were also made for compounds 1a (Table S1) and 2.

Q: The manuscript should include an informative arrow diagram for each of the new compounds. The SI should include a fully tabulated account of all the 2D NMR data for each compound.

A: Not accepted lead drawings with 2D correlations for all isolated compounds, as this greatly complicates the perception of the article. We provided a Figure 4 with NOE correlations for compound 7 to clarify its absolute configuration. We provided the Tables S2 and S3 with 2D NMR data for piltunines A‒F.

Q: Table 5. I am at a loss to understand the significance of this assay. Are the compounds NOs inhibitors? Are they NO quenching agents? Do they block LPS mediated signaling?

A: We have included some data on ROS and NO activity in the text of the manuscript.

The choice of biological tests was determined both by the amount of isolated compounds and the set of currently available test systems. The inhibitory effects of compounds 1-3, 8, 9 on urease activity was also studied. None of the tested compounds exhibited inhibitory effects on the urease enzyme (data don’t shown in manuscript). We added IC50 and positive control data in Table 3. Colony formation assay data (Figure 4) were excluded from the text of manuscript.

Reviewer 2 Report

In the manuscript, the authors reported the isolation of six new sesquiterpenes from marine-derived fungus Penicillium pitunense. Structural analyses of the isolated compounds draw following concerns that should be addressed.

During the structural elucidation of compound 1, the authors placed hydroxy group on C15 in the proposed structure. Deduced from the observed correlations, it is also possible that C14-hydroxylated structure. How did authors distinguish two possibilities? The same question should also be addressed for compound 2.

What is the meaning of the words “biogenetic considerations”? The authors repeatedly used these words for elucidation of the absolute configurations of isolated compounds, but the meaning is not clear in the manuscript. To discuss the absolute configuration of the isolated compounds, experimental ECD spectra should be compared with theoretical ones.

Specific rotation of compound 7 should be compared with that reported in the ref. 11, using the same solvent in the literature. Spectral data for known compounds (8 and 9) should also be added in the supporting information.

Author Response

Dear Reviewer. Thank you for your careful analysis of our paper and valuable comments. Bellow you can find the answers for your questions and comments.

Q: During the structural elucidation of compound 1, the authors placed hydroxy group on C15 in the proposed structure. Deduced from the observed correlations, it is also possible that C14-hydroxylated structure. How did authors distinguish two possibilities? The same question should also be addressed for compound 2.

A: A careful analysis of the observed NOE correlations shows that the hydroxy group in 1 is at C-14, and the 10,11 double bond has the E configuration. Accordingly, we have made changes in the sentence on line 85, Figures 1, 2 and 3 and Tables 1, 2. Similar corrections were also made for compounds 1a (Table S1) and 2.

Q: What is the meaning of the words “biogenetic considerations”? The authors repeatedly used these words for elucidation of the absolute configurations of isolated compounds, but the meaning is not clear in the manuscript.

A: When we used the term “biogenetic consideration” in the text, we mean biogenesis. It is usual term for such case (Kolesnikova S.A. et al. // J. Nat. Prod. 2013. V. 76, No. 9. P. 1746-1752; Zhao Y. et al. // Tetrahedron. 2015. V. 71, No. 18. P. 2708-2718; Yurchenko A. et al. // Marine Drugs. 2019, 17 (10), No 579, DOI: 10.3390/md17100579)

Q: To discuss the absolute configuration of the isolated compounds, experimental ECD spectra should be compared with theoretical ones.

A: The original text of our paper does not contain results of quantum-chemical calculations of ECD spectra. But, in reality, a very extensive theoretical investigation of this property was performed by us for stereoisomers of 1 and 7. We explored quantum chemistry of consequently growing quality: B3LYP_PCM/6-31G(d); B3LYP_PCM/6-311G(d); B3LYP_PCM/6-311++G(d); B3LYP_PCM/cc-pvTz.

The standard overall conventional theoretical scheme (further “standard model”) was used – 1) first, the extended conformational analysis was made for possible stereoisomers of compounds 1, 7; 2) conformations, for which Gibbs free energies are in the range 0 £ DGim £ 3 kcal/mol (where index “m” denotes the most stable conformation) were then selected for calculations of ECD spectra using the TDDFT_PCM level of theory. Obtained results were then statistically averaged.

We found, that for these compounds the “standard model” is insufficient. The main problem concerns the long-wave region of the UV spectra – there is significant absorbance at l » 250 and just at l » 270 nm for 1 and 7 in the experimental UV spectra. But, the energy of the lowest electronic transition, calculated even at the highest level of theory, is l » 215 nm (for 7) and l » 230 nm for (1).

These results pointed out, that any choice of the so-called “UV-shift” DEexcitation = Etheoetical – Eexpeimental, which commonly is used for improving theoretical results, can not be obtained correctly. And that, hence, absolute configurations, obtained based on this choice of the UV shift, will be speculative.

The further extension of theory’s quality – by using augmented basis sets or by using other functionals for DFT should not be looked at this situation as A GOOD design, because THERE ARE MANY OTHER non-considered FACTORS, WHIch THE “STANDARD MODEL” IGNORS COMPLETELY.

For example, all UV/ECD experiments are performed at room temperatures. As a result, all experimentally available contours are averaged over vibrational motions of compounds under study and over the reorganization of the (at minimum) first solvation shell. At present these effects are ignored in theoretical models at all due to the fact, that such investigations are very-very CPU time consuming.

Thus, compounds 1 and 7 are structurally flexible. The seven-membered ring may exist in two conformations. The inversion of five-membered ring (for 7) is strongly coupled with the internal rotation of the vinyl and hydroxyl groups attached to C-10. The internal rotations of OH and COOH groups attached to C-3 are not free, but are coupled with transformations in first solvation shell, constructed with the proton-donor molecules (methanol).

The investigation of these factors and their influences on calculated ECD spectra is a core of the separate publications. In this work we attempted to establish absolute configurations based on detailed analysis of our NMR data and known absolute configuration of the “core molecule” – aspterric acid, for which the absolute configuration was established using X-ray analysis [13].

Q: Specific rotation of compound 7 should be compared with that reported in the ref. 11, using the same solvent in the literature. Spectral data for known compounds (8 and 9) should also be added in the supporting information.

A: Specific rotation of compound 7 was measured in EtOH, as well as in literature ([α]D20 ‒36.2 (c 0.16 EtOH)). Spectral data for known compounds (8 and 9) were added to the SI (Figures S62-S65).

Reviewer 3 Report

In this manuscript, the authors present the isolation of 9 compounds (7 new) from a marine-derived fungus of the Penicillium genus. The biological activity of some of the isolated compounds was checked.

The new compounds were isolated and characterized using standard techniques, which are well explained in the text.

The manuscript is well written, and the contents are of interest to researchers on natural products.

However, there are some issues that have to be addressed for this manuscript to be publishable.

Compound 1 is depicted as having a Z configuration of the exocyclic double bond, but neither in the text or in the experimental part in indicated how was this configuration established. It should be possible using data from the NOESY spectra of 1, 1a. Compound 2 is the acetate of 1, the configuration of the double bond could be determined in this compound and then a correlation with 1 could be established. In any case, this issue should be clarified.

Line 77, the bridge between carbons 2 and 13 is an ether bridge, no an ester one.

In Tables 1 and 2, the units of the spectrometer frequency in the footer are missing. In the header of Table 2 13C is also missing.

Author Response

Dear Reviewer. Thank you for your careful analysis of our paper and valuable comments. Bellow you can find the answers for your questions and comments.

Q: Compound 1 is depicted as having a Z configuration of the exocyclic double bond, but neither in the text or in the experimental part in indicated how was this configuration established. It should be possible using data from the NOESY spectra of 1, 1a. Compound 2 is the acetate of 1, the configuration of the double bond could be determined in this compound and then a correlation with 1 could be established. In any case, this issue should be clarified.

A: A careful analysis of the observed NOE correlations shows that the hydroxy group in 1 is at C-14, and the 10,11 double bond has the E configuration. Accordingly, we have made changes in the sentence on line 85, Figures 1, 2 and 3 and Tables 1, 2. Similar corrections were also made for compounds 1a (Table S1) and 2.

Q: Line 77, the bridge between carbons 2 and 13 is an ether bridge, no an ester one.

A: It was corrected

Q: In Tables 1 and 2, the units of the spectrometer frequency in the footer are missing. In the header of Table 2 13C is also missing.

A: It was corrected

Reviewer 4 Report

The manuscript entitled “Piltunines A-F fro the marine-derived fungus Penicillium piltunense KMM4668” submitted by Shamil Afiyatullov et al. describes the isolation, characterization of piltunines A-F and other known compounds for their biological activity. Please see comments below:

In Figure 1, please include the structure 10, if not revise accordingly. In figure 3, the authors have shown the lone a pairs of electron for representation in structures of 1 and 3, please remove them as they are not necessary to show. Also, please distinguish the double bond from single bond in the structures shown. Please provide 13C DEPT 135 to show the clear differences for the additional confirmation, Include them in the supporting information. In line 103, please discuss how CD analysis helps to distinguish piltunine A from piltunine B along with other new compounds? Please enclose UV chromatograms and IR, elemental analysis for all the new compounds. In addition, please provide chiral HPLC of the isolated piltunines A-F. Please include viability of compound 3-6 in table 3 for comparison and discuss them. In line 177, there is no inhibition of compound 9 when compared to control? Explain. Since the piltunine A (1) is commercially available, please compare your isolated piltunine A for its identity. In the supporting information, please reaquire clear proton NMRs of figure S10, figure S12, figure S25, figure S35 and S45. Please choose the appropriate solvents for their solubility.

Major revisions required for this manuscript.

Author Response

Dear Reviewer. Thank you for your careful analysis of our paper and valuable comments. Bellow you can find the answers for your questions and comments.

Q: In Figure 1, please include the structure 10, if not revise accordingly

A: Figure 1. We changed “Chemical structures of 110” to “Chemical structures of 19

Q: In figure 3, the authors have shown the lone a pairs of electron for representation in structures of 1 and 3, please remove them as they are not necessary to show. Also, please distinguish the double bond from single bond in the structures shown.

A: We removed the lone a pairs of electron for representation in structures 1 and 3. Structures 1 and 3 were deployed so that hybridization of carbon atoms was visible.

Q: Please provide 13C DEPT 135 to show the clear differences for the additional confirmation, Include them in the supporting information.

A: We provided DEPT-135 spectra in SI

Q: In line 103, please discuss how CD analysis helps to distinguish piltunine A from piltunine B along with other new compounds?

A: We did not use CD spectra to compare Piltunines A and B. CD spectra were used to determine the absolute configuration of chiral centers. However, the difference in the intensity of the Coton effect at λ207 does not allow to correctly establish the absolute configuration of Piltunine B. Therefore, we replaced the sentences “The relative configuration of 2 was elucidated based on the observed NOE correlations (Figure S24). Compound 2 showed characteristic Cotton effects (CEs) at λ209+0.46and λ225‒0.51inthe CD spectrum (Figure S2), which were in line with the results for piltunine A (1) and aspterric acid. Thus, the absolute configuration of 2 was established as 2R,3R,6S,7S.” to “The absolute configurations of the chiral centers in 2 were elucidated as 2R,3R,6S,7S based on NOESY data (Figure S27, Table S2) and biogenetic considerations.”

Q: Please enclose UV chromatograms and IR, elemental analysis for all the new compounds.

A: We included UV spectral data of compounds 1‒4, IR spectra in CDCl3 for compound 1, 2, 4, 6 in Experimental. For compounds 3 and 5, it was not possible to record IR spectra due to their limited solubility in chloroform. In the manuscript, we provide HR ESIMS data that establish the elemental composition of the molecule. This is a common practice when analyzing natural compounds and more accurate than elemental analysis.

Q: In addition, please provide chiral HPLC of the isolated piltunines A-F.

A: For what? The purity of the compounds 1–9 is confirmed by NMR, HR ESIMS spectra and optical rotation and cannot be doubted.

Q: Please include viability of compound 3-6 in table 3 for comparison and discuss them.

A: Compounds 1, 2, 7‒9 were isolated first and cytotoxic activity was studied for them. Compounds 3 and 4 were isolated later in small amounts (3 and 1 mg, respectively). For compound 3 an attempt to obtain chiral derivatives using the modified Mosher method was made. It is not currently possible to send samples of compounds 5 and 6 to Germany for studying of their cytotoxic activity.

Q: In line 177, there is no inhibition of compound 9 when compared to control?

A: Colony formation assay data (Figure 4) were excluded from the text of manuscript.

Q: Explain. Since the piltunine A (1) is commercially available, please compare your isolated piltunine A for its identity.

A: It is not possible to buy a commercially available piltunine A for the foreseeable future.

Q: In the supporting information, please reaquire clear proton NMRs of figure S10, figure S12, figure S25, figure S35 and S45. Please choose the appropriate solvents for their solubility.

A: It was corrected

Round 2

Reviewer 1 Report

That the authors revised the previously assigned structures for 1, 1a and 2 makes my point about the need to fully tabulate the 1D and 2D NMR data, and provide arrow diagrams. That they elected not to include arrow diagrams for all the new compounds is unfortunate, but it is their call (it would not be mine).

Cytotoxicity assay: I note the authors have included data for the positive control docetaxel, however, they have not described a negative control (i.e. blank media with a fixed, low conc of the carrier DMSO)?

Colony forming assay: I note that the authors have removed the colony forming assay data (i.e. formerly Figure 4).

LPS/ROS assay - my concerns about this assay have not been attended to, and still stand.

While I still have doubts about the significance of the bioassays, the authors clearly have a different view. I respect this right, and do not wish to stand in the way of publication.

Author Response

Dear Reviewer. Thank you for your careful analysis of our paper and valuable comments. Bellow you can find the answers for your questions and comments.

 Q: That the authors revised the previously assigned structures for 1, 1a and 2 makes my point about the need to fully tabulate the 1D and 2D NMR data, and provide arrow diagrams. That they elected not to include arrow diagrams for all the new compounds is unfortunate, but it is their call (it would not be mine).

A: We still believe that including arrow diagrams for all the new compounds is unnecessary. We previded the data of 1D and 2D NMR spectra for all new compounds in Tables S2 and S3 in SI. For example, configurations of side chains in compounds 3, 5, and 6 follow from the NOE correlations given in these tables.

 Q: Cytotoxicity assay: I note the authors have included data for the positive control docetaxel, however, they have not described a negative control (i.e. blank media with a fixed, low conc of the carrier DMSO)?

A: We have included a negative control proposal “The vehicle (DMSO) treated cells were used as a negative control” in the Experimental section of the manuscript, Line 299.

Q: LPS/ROS assay - my concerns about this assay have not been attended to, and still stand.

A: These methods are standard for finding and evaluating the effectiveness of anti-inflammatory agents.

Ivanchina, N.V.; Kicha, A.A.; Malyarenko, T.V.; Kalinovsky, A.I.; Menchinskaya, E.S.; Pislyagin, E.A.; Dmitrenok, P.S. The influence on LPS-induced ROS formation in macrophages of capelloside A, a new steroid glycoside from the starfish Ogmastercapella. Nat. Prod. Commun. 2015, 10, 1937–1940.

Afiyatullov, S.S.; E.V. Leshchenko, M.P. Sobolevskaya, V.A. Denisenko, N.N. Kirichuk, Y.V. Khudyakova, T.P.T. Hoai, P.S. Dmitrenok, E.S. Menchinskaya, E.A. Pislyagin, D.V. Berdyshev New eudesmane sesquiterpenes from the marine-derived fungus Penicillium thomii Phytochem. Lett. 2015, 14, 209 - 214.

Fedoreyev, S.A.; Krylova, N.V.; Mishchenko, N.P.; Vasileva, E.A.; Pislyagin, E.A.; Iunikhina, O.V.; Lavrov, V.F.; Svitich O.A.; Ebralidze, L.K.; Leonova, G.N. Antiviral and Antioxidant Properties of Echinochrome A. Marine Drugs, 2018, 16 (12), 509. doi: 10.3390/md16120509.

Reviewer 2 Report

The authors addressed questions raised by the reviewers and the manuscript was appropriately revised.

Author Response

Dear Reviewer. Thank you for your careful analysis of our paper and valuable comments. Bellow you can find the answers for your questions and comments.

Reviewer 4 Report

Minor comments:

1) Please correct and show the double bonds of the structures 1 and 3 in figure 3. Did not see any difference from the previous version in terms of double bond.

2) Please label table 2 as 13C NMR data.

3) In table 3, the authors are claiming that compound 8 had high SI but structurewise it is different than the rest of the piltunines A-F. However, IC50s of compound 3, 5 and 6 are missing. it would be great to include for comparison.

Q: Please enclose UV chromatograms and IR, elemental analysis for all the new compounds.

A: We included UV spectral data of compounds 1‒4, IR spectra in CDCl3 for compound 1, 2, 4, 6 in Experimental. For compounds 3 and 5, it was not possible to record IR spectra due to their limited solubility in chloroform. In the manuscript, we provide HR ESIMS data that establish the elemental composition of the molecule. This is a common practice when analyzing natural compounds and more accurate than elemental analysis.

Did not see any UV and IR spectras in the SI.

Author Response

Dear Reviewer. Thank you for your careful analysis of our paper and valuable comments. Bellow you can find the answers for your questions and comments.

Q:. Please label table 2 as 13C NMR data.

A: It was corrected

Q: Please correct and show the double bonds of the structures 1 and 3 in figure 3. Did not see any difference from the previous version in terms of double bond.

A: For the image of 3D models, we use a licensed program Chem3D that takes into account bond lengths, hybridization, angles, etc. The image of double bonds is not necessary, since the figure takes into account the hybridization of carbon atoms. This is a common practice when the structures of natural compounds were shown:

Xing, C.P.; Xie, C.L.; Xia, J.M.; Liu, Q.M.; Lin, W.X.; Ye, D.Z.; Liu, G.M.; Yang, X.M. Penigrisacides A‒D, four new sesquiterpenes from the deep-sea-derived Penicillium griseofulvum. Mar. Drugs 2019, 17, 507‒514. (Figure 2)

Wang, Z.R.; Li, G.; Ji, L.X.; Wang, H.H.; Gao, H.; Peng, X.P.; Lou, H.X. Induced production of steroids by co-cultivation of two endophytes from Mahonia fortune. Steroids 2019, 145, 1-4 (Figure 5)

Lyakhova E. G., Kolesnikova S. A., Kalinovsky A. I., Berdyshev D. V., Pislyagin E. A., Kuzmich A. S., Popov R. S., Dmitrenok P. S., Makarieva T. N., Stonik V. A. Lissodendoric acids A and B, manzamine-related alkaloids from the Far Eastern sponge Lissodendoryx florida. Organic Letters. 2017, 19 (19), 5320–5323. (Figure 3)

Q: In table 3, the authors are claiming that compound 8 had high SI but structure ewise it is different than the rest of the piltunines A-F. However, IC50s of compound 3, 5 and 6 are missing. it would be great to include for comparison.

A: Compounds 1, 2, 7‒9 were isolated first and cytotoxic activity (including IC50s) was studied for them. Compounds 3 and 4 were isolated later in small amounts (3 and 1 mg, respectively). An attempt to obtain chiral derivatives for compound 3 (2 mg) using the modified Mosher method was made. It is not currently possible to send samples of compounds 3, 5 and 6 to Germany for studying of their cytotoxic activity.

Q:. Did not see any UV and IR spectra in the SI.

A: We provided UV and IR spectra in the SI.